# Memorization Without Overfitting: Analyzing the Training Dynamics of Large Language Models

**Kushal Tirumala**[*]    **Aram H. Markosyan**[*]    **Luke Zettlemoyer**    **Armen Aghajanyan**

**Meta AI Research**
{ktirumala,amarkos,lsz,armenag}@fb.com

## Abstract

Despite their wide adoption, the underlying training and memorization dynamics of very large language models is not well understood. We empirically study exact memorization in causal and masked language modeling, across model sizes and throughout the training process. We measure the effects of dataset size, learning rate, and model size on memorization, finding that larger language models memorize training data faster across all settings. Surprisingly, we show that larger models can memorize a larger portion of the data before over-fitting and tend to forget less throughout the training process. We also analyze the memorization dynamics of different parts of speech and find that models memorize nouns and numbers first; we hypothesize and provide empirical evidence that nouns and numbers act as a unique identifier for memorizing individual training examples. Together, these findings present another piece of the broader puzzle of trying to understand what actually improves as models get bigger.

## 1 Introduction

The rate and extent to which a model memorizes its training data are key statistics that provide evidence about how it is likely to generalize to new test instances. Classical frameworks, such as bias-variance tradeoff [31], argued for fitting a training set without full memorization. However, recent work has established a more symbiotic relationship between memorization and generalization in deep learning [13, 26, 28]. This paper empirically studies memorization in causal and masked language modeling, across model sizes and throughout the training process.

Much of the recent performance gains for language models have come from scale, with the most recent models reaching up to $10^{11}$ parameters [22, 73, 83]. Larger models are also known to memorize more training data [16], which is a crucial component of their improved generalization. However, perhaps surprisingly, relatively little work has been done in understanding the impact of scale on the dynamics of language model memorization over training. Existing work focuses on analyzing memorization post-training [16, 47, 88, 95]. In this work, we study the memorization and forgetting dynamics in language models, with a focus on better measuring how they change as we scale up model size. Our primary contributions include:

1. We measure the dependence of memorization dynamics over training on model size (and other factors such as dataset size, overfitting, and learning rate). We find that larger language models memorize training data faster (§ 4).

2. We design controlled experiments that allow us to characterize the forgetting curves in language models (i.e., how language models naturally forget memories throughout training).

---

[*]Equal Contribution

36th Conference on Neural Information Processing Systems (NeurIPS 2022).

Our empirical studies show that forgetting curves have lower bounds — we coin this as the *forgetting baseline* — and that this baseline increases with model scale, i.e., increasing model scale mitigates forgetting (§ 5).

3. We analyze the rates of memorization of different parts of speech, finding that nouns and numbers are memorized much more quickly than other parts of speech (§ 4.4). We hypothesize this is because the set of nouns and numbers can be seen as a unique identifier for a particular sample. We provide evidence to this hypothesis by analyzing the rates of memorization in the setting of an existing unique identifier (§ 4.3).

Together, these findings present another piece of the broader puzzle of trying to understand the unique training dynamics that emerge as models grow in size.

## 2 Background and Related Work

**Memorization in Language Models**: Unintended memorization is a known challenge for language models [14, 85], which makes them open to extraction attacks [15, 89] and membership inference attacks [41, 64], although there has been work on mitigating these vulnerabilities [51, 88]. Recent work has argued that memorization is not exclusively harmful, and can be crucial for certain types of generalization (e.g., on QA tasks) [11, 46, 87], while also allowing the models to encode significant amounts of world or factual knowledge [4, 35, 71]. There is also a growing body of work analyzing fundamental properties of memorization in language models [16, 47, 60, 95]. Most related to our work Carlini et al. [16] analyzes memorization of fully trained language models and observes a dependence on model scale, training data duplication, and prompting context length. While we also study scaling behavior, our focus instead is on the memorization dynamics throughout training.

**Language Model Training Dynamics**: Previous work has extensively analyzed training dynamics to understand how neural models acquire information over training [1, 30, 34, 66, 74]. Saphra and Lopez [80] were the first to analyze training dynamics for language modeling, focusing on the evolution of internal representations over pre-training. This inspired a line of work analyzing how neural language models learn linguistic structure/world knowledge [20, 21, 53], individual words [17], and cross-lingual structure [10] over pre-training. This analysis has been extended to many downstream tasks, including text summarization [33], machine/speech translation [81, 86, 92], and various NLP tasks [36, 61].

**Forgetting in Language Models**: There has also been work studying memory degradation (forgetting) in language models. *Catastrophic forgetting* or *catastrophic interference*, first reported in [59, 77], studies how neural networks tend to forget the information from previous trained tasks or training batches, when trained on new data. This provides a key challenge for continual learning (or life-long learning) [19], where the goal is to gradually learn from a single pass over a, typically very large, stream of data. A number of mechanisms have been proposed for increasing robustness against catastrophic forgetting [2, 18, 24, 49, 58, 82]. There is also a growing body of work demonstrating that both model and dataset scale can make models more resistant to forgetting [65, 75], as well as work characterizing how forgetting naturally occurs in image classifiers [90] and how forgetting can improve training efficiency [5]. *Machine unlearning* is a technique that forces a trained model to forget a previously learned sample [12, 54], which is primarily motivated by data protection and privacy regulations [37, 57, 78, 91]. Our work is unique in its focus on measuring forgetting during training, and quantifying how it varies with scale.

**Scaling Laws**: We have consistently seen performance gains by scaling model size [3, 22, 73, 76, 83], and scale itself has been known to push internal model behavior away from classical bias-variance regimes [67]. Recent efforts have focused on trying to model the scaling laws for language models, including data and model size [44, 79], applications to transfer learning [40], routing networks [23], and various autoregressive generative tasks [39]. While the bulk of work in scaling laws has been empirical, an interesting line of work focuses on theoretically explaining neural scaling laws [8]. Most scaling laws focus only on cross-entropy loss, while we study memorization (defined in § 3).

# 3 Experimental Setup

In order to perform a large-scale study of the dynamics of memorization over training, our memorization metric must be reasonably easy to compute but also precise enough to tell us how much the model will actually remember from the training data. Label memorization [72, 94] [2] is an ideal candidate, because it has consistently provided theoretical insight into underlying properties of neural networks, remains applicable in empirical settings, and is relatively cheap to compute. We formulate our metric as an analog of label memorization for self-supervised settings.

**Definition 1** *Let $V$ denote the vocabulary size. Let $C$ denote a set of contexts, which can be thought of as a list of tuples $(s, y)$ where $s$ is an input context (incomplete block of text) and $y$ is the index of the ground truth token in the vocabulary that completes the block of text. Let $S$ denote the set of input contexts, and let $f : S \rightarrow \mathbb{R}^V$ denote a language model. A context $c = (s, y) \in C$ is memorized if $\mathrm{argmax}(f(s)) = y$.*

Note that a single word can appear as the ground-truth token for multiple contexts. For a given set of contexts $C$ (i.e a given training dataset), we can then analyze the proportion of memorized contexts

$$M(f) = \frac{\sum_{(s,y)\in C} \mathbb{1}\{\mathrm{argmax}(f(s)) = y\}}{|C|}$$

We refer to this as *exact memorization*, although it can also be seen as accuracy since we measure how often the argmax of the language model matches the ground truth token. Throughout this work, when we refer to memorization, we will be referring to Definition 1 unless we specify otherwise.

We define $\tau$ to be a threshold value for $M(f)$, and denote $T(N, \tau)$ as the minimal number of times a language model $f$ with $N$ parameter needs to see each training datapoint in order to satisfy $M(f) \geq \tau$. When leveraging bigger datasets, models are unable to train for multiple epochs, so we instead consider memorization on a per-update basis. We introduce $M_{update}(f, U)$ as the memorization on the batch of data on which the model performs the $U$'th gradient descent update, and define $T_{update}(N, \tau)$ as the minimal number of gradient descent updates a language model with $N$ parameters needs to perform, to satisfy $M_{update}(f, U) \geq \tau$.

Previous work analyzing language modeling memorization defines memorization differently. Motivated by privacy concerns, both [15] and [16] define memorization from a training data extraction standpoint, in which a string $s$ is extractable if it can be produced by interacting with the language model. More specifically, [15] defines a string $s$ as being $k$-eidetic memorized if it is extractable and appears in at most $k$ training examples. [16] defines a string $s$ as $k$-memorized if the language model can produce it via prompting with $k$ tokens of context from *training data*. This definition only works for causal language modeling because of the dependence on prompting with training data; for masked language modeling [16] uses Definition 1 above. Note that if an example is exactly memorized, it is extractable by definition. In other words, both the set of $k$-eidetic memorized tokens and the set of $k$-memorized tokens contain the set of exactly memorized tokens (formally, different exactly memorized tokens may be contained in different sets, depending on $k$). Therefore, analyzing exact memorization gives a type of lower bound on the $k$-eidetic memorization and $k$-memorization. In a different line of work motivated by estimating the influence of individual training examples, [95] defines a training example $x$ as memorized if the difference in expected model performance (where model performance is defined as $M(f)$ above) over subsets of data including $x$ and subsets of data not including $x$, is sufficiently large. This definition pulls from previous work in theoretically analyzing label memorization in classification settings [27].

*Model Architectures*: We replicate publicly available references for Transformer language model architectures [7, 96]. We use the 125M, 355M, 1.3B, 2.7B, 6.7B, and 13B model configurations (see § A.4 for more architectural and training details). We study both causal and masked language models. We train using the FairSeq framework [69] with PyTorch [70] as the underlying framework. For our larger models, we use the fully sharded data-parallel implementation available in FairScale [9] and use Aim experiment tracking [6].

*Datasets*: We use two existing datasets across all our experiments: the WIKITEXT-103 benchmark containing around 103 million tokens [62], and the RoBERTa corpus [55] used to train the original

---

[2]Label memorization in these prior works usually refers to perfectly fitting a given set of labels

RoBERTa model, containing around 39 billion tokens (we refer to this as the ROBERTA dataset). We use both datasets in section 4, and primarily use WIKITEXT-103 in other sections due to computational restrictions.

## 4 Larger Language Models Memorize Faster

Larger neural language models are known to be more sample efficient and require fewer optimization steps to reach the same performance [44] while also converging faster [52], where performance is usually defined as *test* perplexity. In this section, we study $T(N, \tau)$ on the *training* set as a function of $N$ to answer this question.

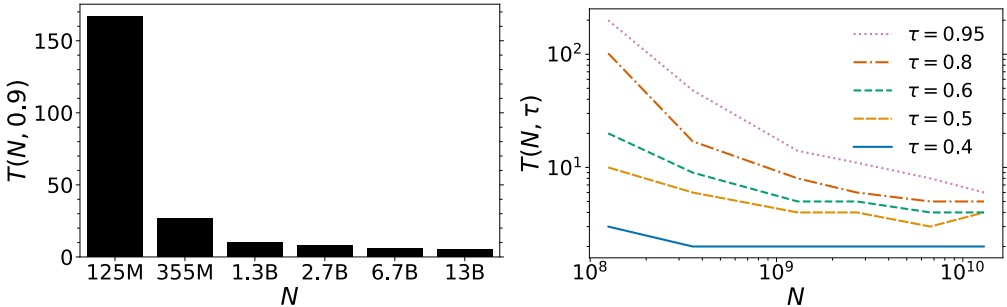

Figure 1: We show $T(N, \tau)$, which is the number of times a language model needs to see each training example before memorizing $\tau$ fraction of the training data, as a function of model size $N$. Result are for causal language modeling on WIKITEXT103, right plot is on log-log scale. Note that generally larger models memorize faster, regardless of $\tau$.

In the left plot of Figure 1, we fix a memorization threshold $\tau = 0.9$ and examine $T(N, \tau)$ as we increase $N$. The larger language models need to see each training datapoint fewer times to achieve 90% exact memorization of the training set; in other words, $T(N, 0.9)$ is monotonically decreasing in $N$. When we vary $\tau$ between 0.4 and 0.95 in the right plot of Figure 1, we still observe that $T(N, \tau)$ is generally decreasing with $N$.[3] For fixed $N$, $T(N, \tau)$ is increasing in $\tau$, which is expected since memorizing more of the training set requires training the model for more epochs. More interestingly, increasing $\tau$ smoothly transitions $T(N, \tau)$ from constant in $N$, to exponentially decreasing in $N$ (the axes are on a log-log scale).

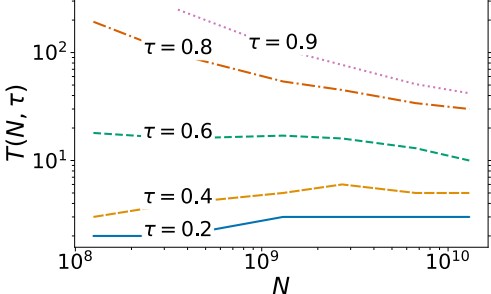

Figure 2: $T(N, \tau)$ as a function of $N$ (shown on log-log scale), for various values of $\tau$ in masked language modeling on WIKITEXT103. We show that larger models initially memorize training data slower, but reach high proportions of training data memorization faster.

---

[3]We fix 0.4 as the lower bound for the range because any lower value for the memorization threshold is achieved within the first few epochs across all model scales (the line in Figure 1 is essentially flat), and 0.95 as the upper bound because higher values require unreasonably long training time for smaller models.

## 4.1 Dependence on Language Modeling Task and Dataset Size

To investigate the dependence of our observations on the particular language modeling task, we repeat this analysis for the masked language modeling task on WIKITEXT103 with mask probability $0.15$. Unlike in causal language modeling, Figure 2 shows that $T(N, \tau)$ is not monotonically decreasing in $N$ for lower values of $\tau$, and is monotonically decreasing in $N$ for higher values of $\tau$, where the phase transition[4] between these two regimes occurs between $\tau = 0.6$ and $\tau = 0.7$. Smaller models memorize the training data quicker initially and slower in the long run (e.g., right plot of Figure 11).

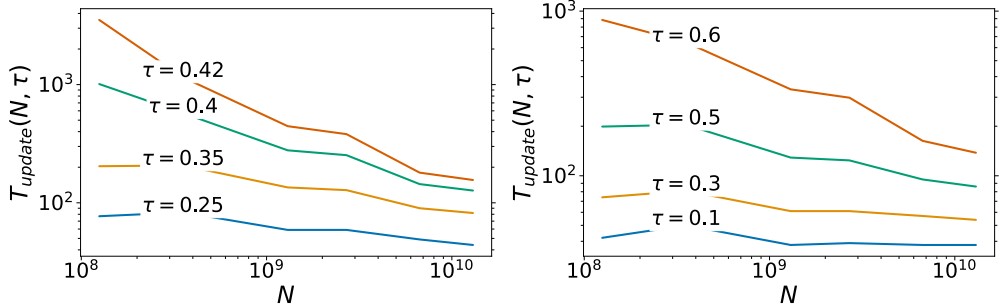

Figure 3: We show $T_{update}(N, \tau)$, which is the number of gradient descent updates $U$ a language model needs to perform before memorizing $\tau$ fraction of the data given on the $U$'th update, as a function of model size $N$. Result are for causal (Left) and masked (Right) language modeling on the ROBERTA dataset, on a log-log scale. We show that larger models memorize faster, regardless of $\tau$.

Language model training is heavily dependent on the dataset size [44], and therefore we expect $M(f)$ to be similarly impacted. In Figure 3, we analyze training set memorization on the much bigger ROBERTA dataset for both masked and causal language modeling. With large datasets such as ROBERTA dataset, it becomes infeasible to perform multiple epochs and evaluate memorization on the entire training set, especially when training larger models. Consequently, we focus on smaller values of $\tau$ and investigate the number of gradient descent updates it takes to reach memorization thresholds, i.e., $T_{update}(N, \tau)$. In Figure 3 we observe a similar trend as Figure 1, where $T_{update}(N, \tau)$ is monotonically decreasing with $N$ for various $\tau$, in both masked and causal language modeling. Unlike with WIKITEXT103, masked language modeling does not have a phase transition for $\tau$.

## 4.2 Why Do Larger Models Memorize Faster?

A natural question at this point is to ask why larger models memorize faster? Typically, memorization is associated with overfitting, which offers a potentially simple explanation. In order to disentangle memorization from overfitting, we examine memorization before overfitting occurs, where we define overfitting occurring as the first epoch when the perplexity of the language model on a validation set increases. Surprisingly, we see in Figure 4 that as we increase the number of parameters, memorization before overfitting generally increases, indicating that overfitting by itself *cannot* completely explain the properties of memorization dynamics as model scale increases.

The learning rate is not constant across our training configurations. Intuitively, larger learning rates should lead to quicker memorization. To investigate to what extent our results can be explained by learning rate, we take a subset of the architectures available above and train on the WIKITEXT103 dataset across a standard range of learning rates while measuring memorization, in Figure 5. Even if we fix a learning rate, larger models reach $0.9$ memorization faster, suggesting that our results are not caused solely by differences in learning rates. Interestingly, sensitivity to learning rate generally decreases as we increase the model size. We also notice in Figure 5 that $T(N, \tau)$ goes down initially (for low LRs) and eventually rises (for high LRs), and as the long as the chosen learning rate places us near the lowest point on the curve, the memorization dynamics do not change significantly (note that axes are on log-scale). This result is consistent with the growing intuition that for neural language models past a particular scale, the learning rate is not a significant hyperparameter [44].

---

[4]"Phase transition" is used in physics to describe significant changes in system behavior that occurs due to varying a parameter, such as temperature. In this case, the parameter is $\tau$

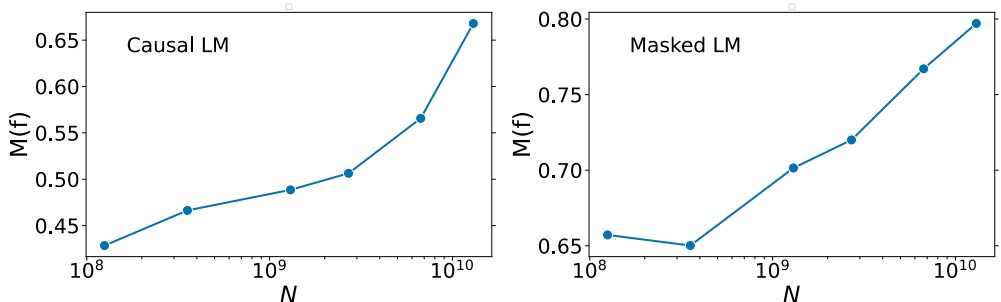

Figure 4: Proportion of training data memorized $M(f)$ before overfitting, as a function of model size $N$ (plotted on a log scale). Results are for causal (left) and masked (right) language modeling on WIKITEXT103. Note that larger models memorize more before overfitting.

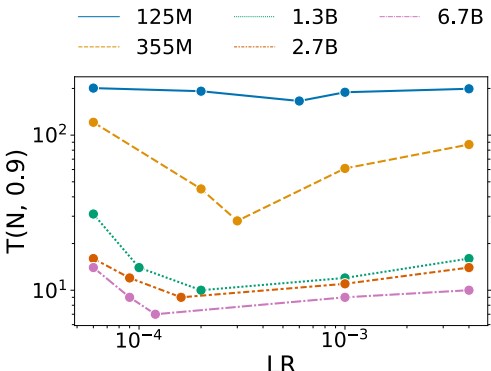

Figure 5: Examining the effect of learning rate (LR) on number of times model needs to see each training example in order to reach 0.9 proportion of training data memorization $T(N, 0.9)$. Each line corresponds to a different model size performing causal language modeling on WIKITEXT103. We demonstrate that larger models memorize faster for a fixed learning rate.

Exhaustively searching all such possible factors is intractable, and providing a complete explanation for why larger models memorize faster is outside the scope of this work. Instead, in the following sections, we present studies that we hope will expand the toolkit for answering such questions.

### 4.3 Memorization via. Unique Identifiers

Recent work studies how to use external memory to improve performance [11, 35, 46, 87]. In this subsection, we question whether such architecture changes are necessary. Motivated by information retrieval systems, we take a simple approach — we prepend a unique identifier to every example in the training set and examine whether memorization speed increases. Specifically, we fix the language modeling task as causal language modeling on WIKITEXT103 with the 125M parameter model, and in front of every training example, we insert the string `document ID <unique_id>` where `unique_id` is a unique integer, one for each training context. In order to utilize all these unique integers, we must add them to the dictionary of tokens, which causes a significant increase in the model size since the last layer in the language model must have an output dimension equal to the size of the dictionary. Therefore, any change in $M(f)$ dynamics could be attributed to the extra parameters we add from increasing dictionary size. To control for this, we first examine the effect of just increasing dictionary size (without using any of the added tokens). Then, we utilize those added tokens to prepend every training example and observe the change in $M(f)$ dynamics. In Figure 6, we see that increasing the dictionary size does improve the speed of memorization. Even though we previously demonstrated that larger models memorize faster, this is still surprising considering that we do not increase parameter size in a significant way — we are effectively adding fake tokens to the dictionary. Moreover, when we leverage those added tokens to identify training examples

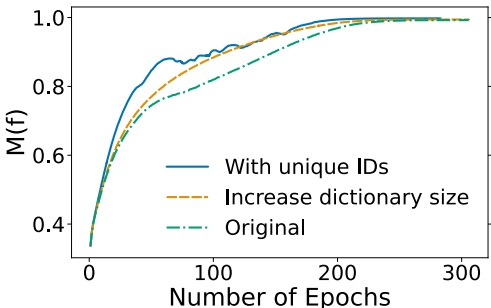

Figure 6: The impact of adding unique identifiers to training examples on memorization $M(f)$ training dynamics for causal language modeling (125M) on WIKITEXT103. The green line is the original 125M model. The orange line is the model after adding unique identifiers to the dictionary (which increases model size). The blue line prepends these unique identifiers for each training example. Note that adding unique identifiers leads to faster memorization of training data.

uniquely, we see yet another gain in memorization, although prompting using a `document ID` shifts memorization dynamics away from being monotonically increasing over time.

## 4.4 Memorization Through the Lens of Parts of Speech

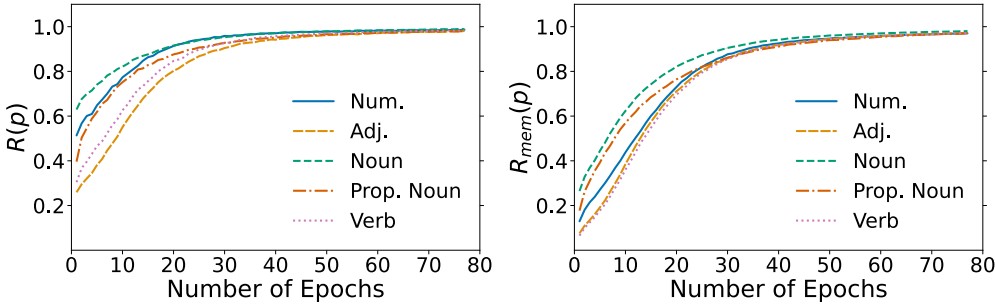

Figure 7: The ratios $R(p)$ (Left) and $R_{mem}(p)$ (Right) over training. $R(p)$ represents proportion of POS correctly memorized (the language model outputs the right POS, but not necessarily the correct word). $R_{mem}(p)$ represents the proportion of exactly memorized tokens for a particular POS $p$. Results are for causal language modeling (355M) on WIKITEXT103. In both plots, we consider numerals, proper nouns, verbs, nouns, and adjectives as potential parts of speech (i.e., values for $p$). We show that nouns and numerals are memorized faster than other parts of speech.

In the previous section, we showed that a unique identifier enhances memorization. Regular text also contains strong proxies to unique identifiers in the form of numerals and proper nouns. Motivated by this, we study syntactic features of memories using part-of-speech (POS) tagging.[5] We track the ratio $R(p)$ of the number of positions for which the part of speech $p$ was correctly predicted to the total number of tokens in the ground truth tagged with that part-of-speech $p$ (left plot in Figure 7). In the right plot of Figure 7 we show a similar ratio, denoted $R_{mem}(p)$, but the numerator only considers the tokens that are also exactly memorized. The correctly predicted part of speech does not necessarily imply exact memorization, which is clearly illustrated by Figure 7 where we see the language model memorizing parts of speech faster than the exact value of the token. While all parts of speech are eventually memorized, *some parts of speech are memorized faster*, which aligns with previous work [20]. However, unlike previous work[6] we find that nouns, proper nouns, and numerals are memorized noticeably faster than verbs and adjectives, both in terms of $R(p)$ and $R_{mem}(p)$. This has potential implications for privacy, since sensitive information is likely to be a noun/proper noun/numeral. Our findings also very loosely align with work studying child language acquisition [29].

---

[5]We use spaCy [42] to identify parts of speech in a text.

[6]This difference could be due to model family (we use causal LMs while previous work uses masked LMs)

## 5 Forgetting Curves in Language Models

This section studies the dual of memorization — forgetting in language models. Inspired by the *forgetting curve* hypothesis, according to which human memory declines over time when there is no attempt to retain it [56], we are interested in understanding the dynamics of memory degradation in language models.

We first choose a batch of data not available in the training set, i.e. a batch of data from a validation set. We refer to this batch of data as the *special batch*. We then take a checkpoint from model training, plug in the special batch so that the model can train on it, and resume standard training on the training set. We then evaluate how memorization degrades on the special batch and analyze the various factors the forgetting curve may depend on. We use the entire validation set as the special batch throughout this section. The special batch is only seen **once** when it is immediately introduced.[7]

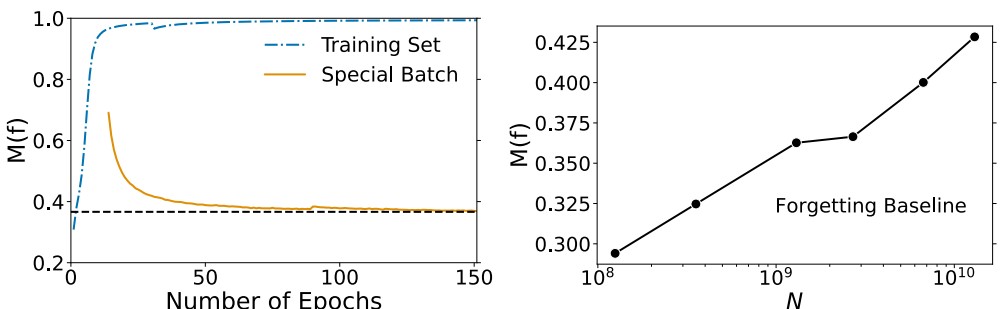

Figure 8: Left: forgetting curve for causal language modeling (2.7B) on WIKITEXT103. The dashed horizontal line indicates the lowest proportion of special batch data memorized throughout training, i.e., the forgetting baseline. Right: forgetting baseline as a function of model size $N$ (plotted on log scale). We show that as model scale increases, the forgetting baseline value increases.

In the left plot of Figure 8, we show the forgetting curve for the 2.7B model. Exact memorization on the special batch degrades quickly at first, but slows down exponentially as we continue training[8] (see Figure 15 in § A.2.2). In other words, the forgetting curve on the special batch seems to approach a baseline — we refer to this trend as the *forgetting baseline*. We approximate the forgetting baseline by looking at the lowest memorization value on the special batch throughout training.

We show the forgetting baseline as a function of the model scale in the right plot of Figure 8. We see that the numerical value for the baseline is monotonically increasing with the model scale. This implies that larger models forget less, aligning with recent work studying catastrophic forgetting on image classification tasks [75]. This is beneficial because larger models can leverage more information from previous tasks; however, from a privacy perspective, this is not ideal because it implies larger models may be potentially retaining more sensitive information from training data.

We also investigate the sensitivity of the forgetting baseline on data batch order. In Figure 9, we perform the same forgetting curve analysis described above but start the analysis at different training checkpoints (we start at the 14th, 39th, and 63rd epochs). This way, we alter the order of the data batches given to the model (since the special batch will appear in a different place in the global order of data batches given to the model) without drastically changing the experimental setup. We observe that the forgetting baseline is not sensitive to data batch order.[9]

---

[7]This experimental setup is different from catastrophic forgetting, as we fix the data distribution by pulling the special batch from the same dataset as the training set. Similarly, it differs from machine unlearning since we are not algorithmically removing information from a language model; instead, we analyze natural forgetting. It is also different from intrinsic hallucination [43], where there is an assumption that contradicted output is semantically correct (e.g., the language model outputs a wrong date).

[8]The average sequential difference in memorization (on the special batch) on the last 3 epochs of training is at most on the order of $10^{-3}$, whereas the average sequential difference in the first 3 epochs of training is consistently on the order of $10^{-2}$.

[9]The max difference between the numerical values for the baseline are on the order of $10^{-3}$

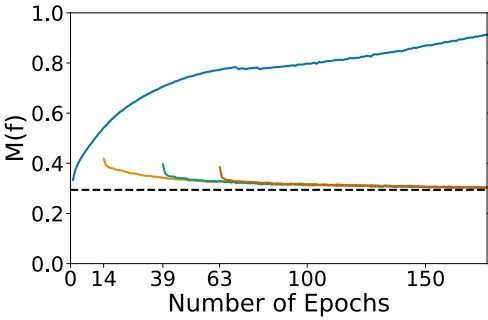

Figure 9: We empirically show that the forgetting baseline does not depend on data batch ordering. We inject the special batch into the training set at the 14th, 39th, and 63rd epochs, and evaluate proportion of special batch data memorized as we continue training. Results are for causal language modeling (125M) on WIKITEXT103.

Motivated by *replay methods* from continual learning (see [24] for a survey) and work in promoting retention memories through *repetition* in both humans [45, 68, 84] and neural models [5], in Figure 10 we study the effect of repetition (left) and spaced repetition (right) on the forgetting baseline. In the left plot, we inject the special batch into the training set multiple times before continuing training on the training set alone. We observe that the forgetting baseline is monotonically increasing as a function of repetition frequency (differences in the baseline value are on the order of $10^{-2}$). To study the spaced repetition, we periodically inject the held-out set into the training set, train on it once, and then continue training on the training set alone. We see in the right plot of Figure 10 that spaced repetition incurs minimal effect on the forgetting baseline (on the order of $10^{-3}$), independent of the length of spacing between the repetitions.

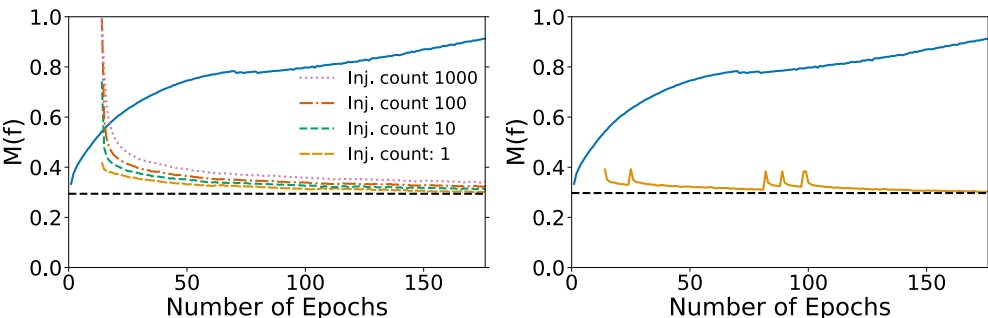

Figure 10: Effect of repeated injection (Left) and spaced repetition (Right) on special batch memorization. Results are for causal language modeling (125M) on WIKITEXT103. The solid upper curve represents the training set memorization. We show that repeated injection increases the forgetting baseline, whereas spaced repetition has minimal effect.

An exciting direction for future work will be to understand the structure of the baseline — for example, understanding what types of tokens (parts of speech, synonyms, facts, syntax) are memorized in the baseline and the overlap of tokens memorized in the baseline with tokens in the training set.

## 6 Conclusions and Discussion

We study the properties of memorization dynamics over language model training and demonstrate that larger models memorize faster. We also measure the properties of forgetting curves and surprisingly find that forgetting reaches a baseline, which again increases with the model scale. Combined with memorization analyses that expose the unintuitive behavior of language models, we hope to motivate considering memorization as a critical metric when increasing language model scale.

Most work studying memorization in language modeling is primarily motivated by privacy (see § 2). While theoretically, there are well-established frameworks to quantify privacy such as differential

privacy [25], empirical privacy in language modeling is not well-defined — does memorizing common knowledge count as information leakage? Does outputting a synonym count as harmful memorization? As per our Definition 1, we implicitly focus on information that is sensitive if outputted verbatim (phone numbers, SSNs, addresses, medical diagnoses, etc.), rather than capturing all aspects of privacy. It is also known that text data used for training language models contain certain biases and stereotypes (e.g., [32]); therefore, our work has similar implications for how long language models can train before they definitively memorize these biases from training data.

We also hope our work highlights the importance of analyzing memorization dynamics as we scale up language models, instead of only reporting cross entropy. Cross-entropy loss and memorization capture different behavior — for example, in many of our memory degradation experiments, even though memorization approaches a baseline, we observe that perplexity is still increasing (see Figure 14 in § A.2 for an example). This implies that the model is becoming unconfident about its exact predictions, which we can only conclude because we inspect both loss and memorization. More importantly, the forgetting baseline behavior would be entirely obscured if we did not inspect memorization dynamics. Similarly, there are multiple instances where we uncover interesting behavior *because* we focus on memorization dynamics (§ 4.4, § 4.3, § A.3), rather than focusing only on cross-entropy loss.

# 7    Acknowledgements

The authors would like to thank Adina Williams, Chuan Guo, Alex Sablayrolles, and Pierre Stock, for helpful discussions throughout the course of this project. The authors would also like to researchers at FAIR who commented on or otherwise supported this project, including Shashank Shekhar, Candace Ross, Rebecca Qian, Dieuwke Hupkes, and Gargi Ghosh.

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
