# A    Appendix

## A.1    Full Memorization Dynamics Over Training

For completeness, in this section we plot our memorization metric $M(f)$ over training for all model sizes. In any of these plots, observe that taking a horizontal slice for a fixed $\tau$ is equivalent to computing $T(N, \tau)$. In Figure 11, we plot $M(f)$ over training for WIKITEXT103. We see that generally (across language modeling tasks and and values of $\tau$), larger models memorize faster. We do notice a caveat in Figure 11, where we observe that in initial stages of training, smaller models memorize faster, but larger models eventually surpass smaller models.

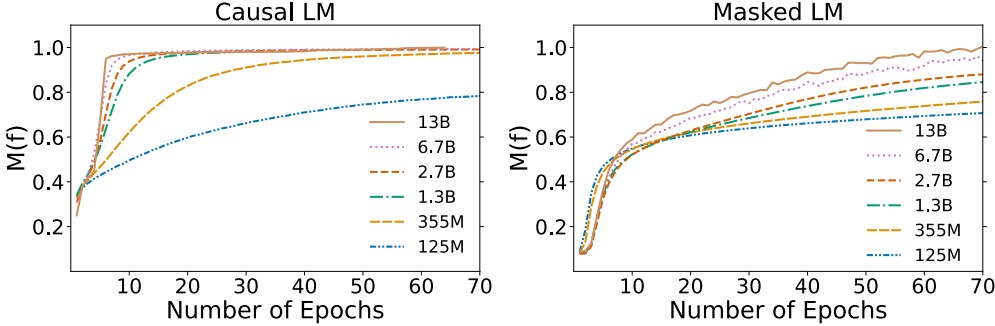

Figure 11: Proportion of training data memorized $M(f)$ over training, for causal (Left) and masked (Right) language modeling on WIKITEXT103. The $x$-axis describes the number of epochs, and $y$-axis denotes $M(f)$ as defined in § 3. Generally, we see that larger models memorize training data faster.

When we analyze larger datasets, performing multiple epochs of training becomes infeasible, and so we track memorization with each gradient descent update. Similarly, we cannot analyze $M(f)$ for the entire training dataset. We use notation introduce in § 1, specifically $M_{update}(f, U)$ where $U$ is the number of gradient updates performed on model $f$. This quantity is defined as the memorization on the batch of data given to the model on the $U$'th update. In figure 12, we take a rolling average with window size 5 when plotting $M_{update}(f, U)$ to smooth out curves.

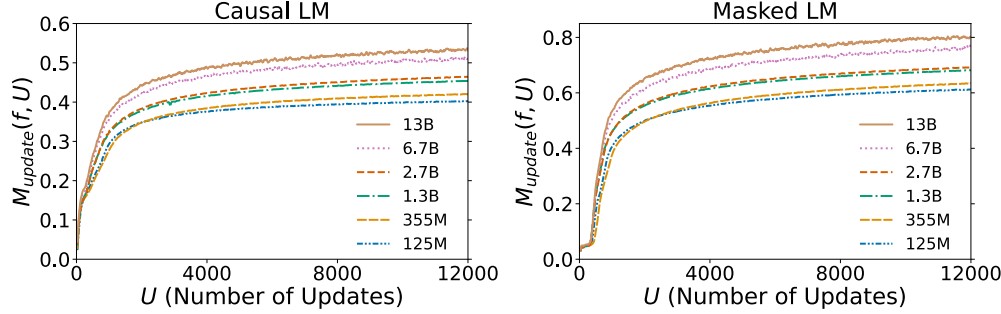

Figure 12: Proportion of training data memorized $M(f)$ over training, for causal (Left) and masked (Right) language modeling on the ROBERTA dataset. The $x$-axis describes the number of gradient descent updates, and the $y$-axis denotes a rolling average (window size 5) of $M_{update}(f)$ as defined above. We again notice that larger models memorize training data faster.

To check that $M_{update}(f, U)$ is a viable proxy for $M(f)$, in Figure 13, we plot both $M(f)$ and $M_{update}(f, U)$ up to 30000 updates for two model sizes. We fix 30000 as the upper bound, because we only train some model sizes up to 30000 updates in the ROBERTA experiments in § 4, and therefore can only completely assess the impact of scale on $M_{update}(f, U)$ dynamics up to 30000 updates. We see that $M_{update}(f, U)$ has periodic behavior, but overall does not deviate too much from $M(f)$.

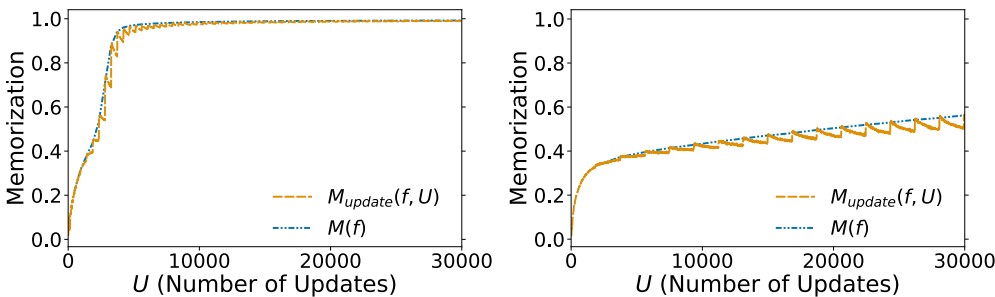

Figure 13: We show training data memorization evaluated at the end of an epoch $M(f)$, and at the end of each gradient descent update $M_{update}(f, U)$, over training. Results shown are for causal language modeling on WIKITEXT103 dataset for 13B (Left) and 125M (Right) model sizes. We note that $M_{update}(f, U)$ closely tracks $M(f)$ throughout training.

### A.1.1 Limitations of Definition 1

We note that Definition 1 is not the best way to study memorization: it ignores model confidence and it does not normalize for duplication in the training set (it is known that duplication in the training set helps models memorize tokens [16, 50]). However, as mentioned in Section 3, all previous definitions of memorization seem to involve Definition 1 in some form. In this way, we study a metric fundamental to memorization regardless of the precise definition of memorization.

## A.2 Forgetting Baseline Analysis

### A.2.1 Perplexity Versus Memorization

This section shows how perplexity and memorization on the special batch evolve over training. In Figure 14 we see that perplexity continues to increase over training, while memorization flatlines. This is a clear experimental setup where we find cross-entropy loss capturing different behavior from memorization. We show plots for the 1.3B model scale, although all of the experiments in § 5 exhibit very similar trends.

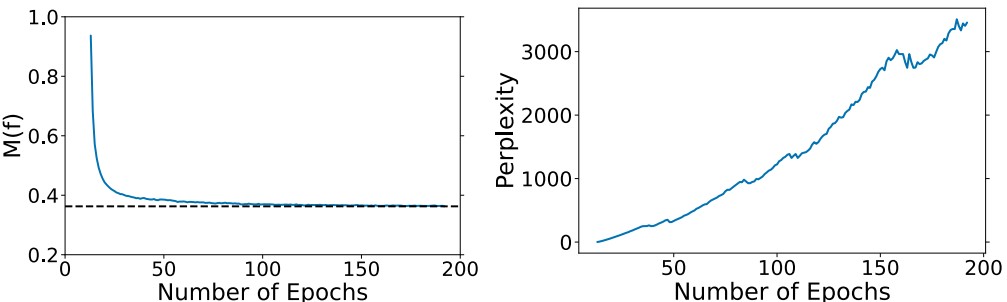

Figure 14: Proportion of special batch data memorized $M(f)$ (Left) and perplexity of special batch (Right) in the forgetting baseline experimental setup described in § 5. Results are for causal language modeling on WIKITEXT103 with 1.3B model size. We notice that memorization of the special batch flattens, while perplexity continues increasing.

### A.2.2 Verifying Existence of Baseline

To verify the existence of the forgetting baseline discussed in § 5, we observe the sequential difference in $M(f)$ of the special batch, from epoch to epoch. More formally, if $M(f)_T$ denotes the memorization at epoch $T$, we investigate $\texttt{diff}(T) = M(f)_T - M(f)_{T-1}$ on the special batch, for $T > 1$. In Figure 15 we show this plot for a few model scales, and we clearly see that the sequential difference in $M(f)$ exponentially approaches 0.

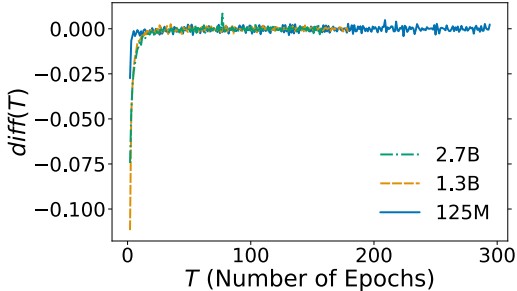

Figure 15: Exploring the sequential difference in proportion of training data memorized $M(f)$ on the special batch over training. The $x$-axis denotes the number of epochs (i.e. $T$) and the $y$-axis denotes the sequential difference in $M(f)$ from the $(T-1)$'th epoch to the $T$'th epoch (i.e $\texttt{diff}(T)$). Results shown are for causal language modeling on WIKITEXT103. We show that sequential difference in memorization exponentially approaches 0.

### A.3 Analyzing Memory Unit Length Over Training

This section investigates a fundamental property of memories — memory unit length $L$. We look at individual tokens memorized as having length $L = 1$, memorized bigrams as having length $L = 2$, memorized trigrams as having length $L = 3$, etc. Analyzing memory length is interesting because it has implications for how language models retain $n$-grams, which are an important part of language. Moreover, recent work shows that chain-of-thought prompting improves language model performance [93]; understanding memory unit length informs us whether a similar method might work for improving performance when training (if a language model has low memory unit length, then including chain-of-thought-type texts in the training set might not have a significant effect). An empirical side note is that these experiments were run separately from the main paper experiments, so we provide original $M(f)$ curves for reference.

We track the average value of $L$ across the entire training dataset for causal language modeling on WIKITEXT103. Note that in our all our experiments, the sequence length is constrained to be less than $512$ tokens, with an average sequence length of $430.12$ on WIKITEXT103. In the left plot of Figure 16 we analyze the average memory unit length over training for two model sizes. We observe across model sizes that average memory unit length steadily increases over time, roughly taking a sigmoidal shape. We notice that the larger 2.7B model has an average $L$ increasing faster than the 125M model. This is consistent with our previous results because we know larger models memorize, and some of these tokens are likely to be adjacent to each other, especially as the model achieves higher values of $M(f)$. Surprisingly, we see that the average memory unit length is much lower than the average sequence length of $430.12$, suggesting that even with high individual token memorization (which is achieved as shown in the right plot of Figure 16), there are always tokens in the middle of a text that the language model has not yet memorized, which break up the memories.

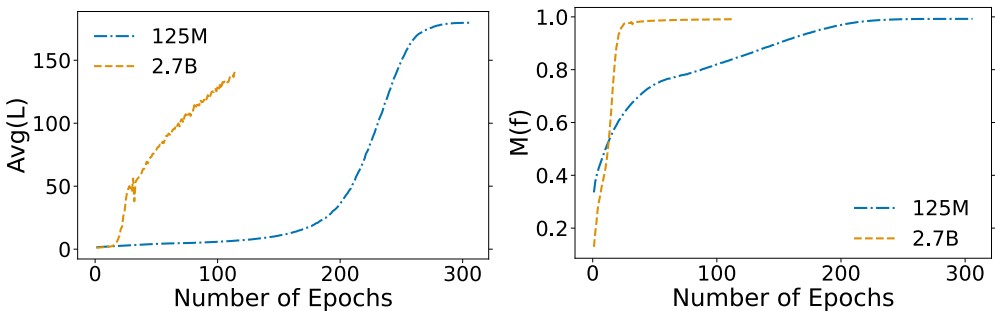

Figure 16: Left: Examining average memory unit length $L$ (averaged over the entire training dataset), as function of number of epochs. As a reference, we show the memorization dynamics $M(f)$ on the right. Results shown are for causal language modeling on WIKITEXT103.

## A.4 Model Training/Dataset Details

In this section, we layout the details of experiments, although most training details we pull directly from publicly available references [7, 96]. As such, we provide the details of model architectures using the same style as Table 1 in [96] for ease of comparison. All models use GELU activation [38] for nonlinearity. We leverage the Adam optimizer [48], with $\beta_1 = 0.9$, $\beta_2 = 0.98$, and $\epsilon = 10^{-8}$. For reproducibility, we set weight decay to 0, dropout to 0, and attention dropout to 0. We use a polynomial learning rate schedule, and following [7, 96] we scale up our learning rate from 0 to the maximum learning rate over $375M$ tokens, and scale down to 0 over the remaining $T - 375M$ tokens (for all masked language modeling experiments, and all ROBERTA experiments, we have $T = 300B$; for causal language modeling experiments on WIKITEXT103 we have $T = 100B$). We fix a sequence length of $512$ across all experiments, but we break input text up into complete sentences, so not all input texts have length exactly equal to $512$. In masked language modeling experiments, we use a mask probability of $0.15$. When training language models, we use the standard procedure of minimizing cross-entropy loss, and use dynamic loss scaling [63].

Table 1: Model architecture details. # L denotes the number of layers, # H denotes the number of attention heads, and $d_{model}$ denotes embedding size. Global batch size denotes the total number of tokens the model processes in a batch of data. Note that most of the values in this table are the same as Table 1 in [96].

| Model Scale | # L | # H | $d_{model}$ | Learning Rate (LR) | Global Batch Size |
|---|---|---|---|---|---|
| 125M | 12 | 12 | 768 | 6.0e-4 | 0.5M |
| 355M | 24 | 16 | 1024 | 3.0e-4 | 0.5M |
| 1.3B | 24 | 32 | 2048 | 2.0e-4 | 1M |
| 2.7B | 32 | 32 | 2560 | 1.6e-4 | 1M |
| 6.7B | 32 | 32 | 4096 | 1.2e-4 | 2M |
| 13B | 40 | 40 | 5120 | 1.0e-4 | 2M |

As mentioned in § 3, we use FairSeq [69] which relies on PyTorch [70]. When training models, we leverage fully sharded data-parallel implementation of models in FairScale [9]. We utilize NVIDIA A100 GPUs with 40GB of memory. Increasing model scale requires different amounts of GPUS: 125M and 355M generally required 16 GPUS, 1.3B required 32 GPUS, and 2.7B, 6.7B, and 13B generally required 64 GPUS (although some experiment runs were launched with 128 GPUS in order to decrease training time). Exact training time varied depended on model scale and dataset size, but all models were trained for up to 140 hours.

In both datasets we use, there is a possibility for sensitive or offensive text to be included in the training set, since both benchmarks use data that is from the Internet. We also note that the WIKITEXT103 benchmark we use throughout the work is available under the Creative Commons Attribution-ShareAlike License. The ROBERTA dataset we use refers to the corpora of text originally used to train the RoBERTa model (see [55]). This dataset not publicly available under any license, however subsets of data that make up the corpus are publicly available.