# OpenReview forum: "Memorization Without Overfitting:  Analyzing the Training Dynamics of Large Language Models"
_NeurIPS.cc/2022/Conference — NeurIPS 2022 Accept_

### Official Review · Reviewer_KjpN · 2022-07-07

**Rating:** 8
**Confidence:** 4
**Soundness:** 3 good
**Presentation:** 3 good
**Contribution:** 3 good

**Summary:**

This paper investigates the memorization and forgetting dynamics of large language models during training, as a function of model and dataset sizes, learning rate, and task (i.e. causal vs masked language modeling). The paper measures memorization as the proportion of correctly predicted labels (which corresponds to predicting the correct next/masked token) during training, and forgetting as the decline in memorization.

The main experimental results show that: (i) larger models memorize data faster and forget less; (ii) memorization occurs before overfitting; and (iii) memorization occurs for both small and large datasets, and for both causal and masked language modeling, regardless of the learning rate. The paper further investigates the memorization mechanism and shows that the presence of unique identifiers in the training data can contribute to memorization, e.g. models tend to memorize numerals, nouns and proper nouns faster than other parts-of-speech. Moreover, the paper shows that while memories from a dataset are forgotten when training with other data, the degree to which forgetting occurs has a lower bound that depends on the model size (larger models having a higher lower bound).

**Questions:**

- Any hypotheses as to why memorization is faster when adding tokens to the dictionary without even using them?


**Limitations:**

authors have discussed the limitations

**Strengths And Weaknesses:**


This paper addresses an important aspect of large language models: their ability to memorize the training data without necessarily overfitting. This unexpected behavior has been observed in previous work and calls into question the classical framework of bias-variance tradeoff. However, little is understood about the dynamics of memorization and forgetting in these models. Yet, this can have problematic implications  for real-world deployments related to e.g., privacy (e.g., models can memorize personal identifiers or other sensitive information) and fairness (e.g., models can memorize negative stereotypes about groups of individuals). This paper gives insights into the memorization and forgetting dynamics in large language model training and about the scaling laws that seem to govern these models.
Overall, the paper is well written and clear and the main claims are well supported by experimental evidence.

---

> ### Author Response · Authors · 2022-08-02
> **RE: Reviewer KjpN**
>
> Thank you for taking the time to review our paper! We are glad you agree with our claims and found our paper clear and well-written.
>
> To answer your question: “Any hypotheses as to why memorization is faster when adding tokens to the dictionary without even using them?”:
>
> Our hypothesis is that adding “fake” tokens to the dictionary increases the number of trainable parameters in the model (in the Embedding layer and the last Linear layer), this allows the model to memorize training data faster. This is supported by the evidence in Section 4 that larger models memorize faster, although in Section 4 we increased # parameters in more conventional ways. Taken together, this seems to point towards the hypothesis that any increase in # parameters (however unconventional) will generally lead to faster memorization of training data. We say “generally” mainly because very recent work by Google/Deepmind has demonstrated that different transformer architectures have different scaling laws [1], indicating that memorization scaling laws will also slightly differ if we change *how* we increase the # of parameters.
>
> ### Citations:
> [1]: Tay, Yi, et al. "Scaling Laws vs Model Architectures: How does Inductive Bias Influence Scaling?." arXiv preprint arXiv:2207.10551 (2022).

---

### Official Review · Reviewer_YUcS · 2022-07-09

**Rating:** 7
**Confidence:** 4
**Soundness:** 2 fair
**Presentation:** 3 good
**Contribution:** 2 fair

**Summary:**

This paper analyzes the phenomenon of memorization by Transformer language models in the light of model size, catastrophic forgetting and memorization of unique tokens such as part of speech. The paper finds that larger the size, more the model is prone to memorize train distribution. Larger models also tends to forget less, and they seem to memorize unique parts of speech tokens such as nouns and numbers.



**Questions:**


- In L122, the authors seem to refer the BookCorpus as "RoBERTa" dataset. Any specific reason why?
- Furthermore, BookCorpus which is used to trained RoBERTa model comes in two sizes: 16GB (which is used for RoBERTa-base) and 160GB (which is used for RoBERTa-large). I'm not sure which one is being used here.
- What is the model family used in these experiments? Are the results specific to Causal language models or Masked Language Models? Do the authors expect the results to differ based on the class of the model family being used?
- In the catastrophic forgetting experiment, the _special batch_ is taken from the validation set of Wiki 103, which is in-distribution with the training set. If I were to choose a special batch being significantly out-of-distribution with respect to the training data, would the same results hold?



**Ethics Review Area:**

["I don’t know"]

**Limitations:**

The authors have not discussed the limitations of their work thoroughly. As I mentioned above, one big limitation is the usage of a significantly small dataset to explain the behaviour of large language models, those of which are typically trained on much larger datasets (for eg, The Pile or C4). This is a window of improvement for the paper, and the authors should try to clearly set the expectations early on for the reader.


**Strengths And Weaknesses:**

- The paper is well written and easy to follow.
- The experimental protocol to measure memorization makes sense, and a good spread of model sizes have been used. It is a bit confusing to understand which model family has been used (as the authors only state they have used "Transformer language model architectures" in L114).
- While I mostly agree with the empirical results, the issue I have with them is that the majority of experiments are conducted on Wiki Text 103, which is 500MB dataset. This dataset is significantly small compared to those being used to train the large models (most of the model sizes considered in this paper are typically trained on significantly larger datasets). Thus, while undoubtedly the models somewhat favor memorization, it is unclear whether they still do the same when the input data is large enough. While I do agree such a study would be quite expensive and perhaps intractable, a disclaimer or limitation of this study should have been explicitly mentioned (on the data size front).
- The study on catastrophic forgetting is quite interesting - it is nice to see a forgetting baseline set by the models of individual sizes.
- The study on parts of speech is especially interesting, however the section lacks a bit on the explanation of the behaviour. Specifically, nouns, proper nouns and numerals being memorized noticeably faster might be due to the frequency effect (Wei, J., Garrette, D., Linzen, T., & Pavlick, E., Frequency Effects on Syntactic Rule Learning in Transformers, arXiv:2109.07020 [cs], 2021). It would be interesting/beneficial for the paper to study this angle further even using Wiki Text 103 dataset, to see if there are any correlation with memorization with the frequency of individual POS class.

*Update*: I upgraded my initial evaluation after reading the author responses.

---

> ### Author Response · Authors · 2022-08-02
> **RE: Reviewer YUcS**
>
> Thank you for thoroughly reviewing our paper! We are glad that you found our work well-written and interesting. We would like to address your comments individually:
>
> ### Weaknesses
> W1: “It is a bit confusing to understand which model family has been used as the authors only state they have used ‘Transformer language model architectures’ in L114”
>
> A: We see how this portion is confusing, as we only specify the broad class of models, and leave the details of the architectures to the cited works in L114. We have updated the draft (L124-125 in the updated draft) to include that we consider both causal and masked language models.
>
> W2: Dataset size issue (we don’t consider large datasets for all our experiments)
>
> A: We completely agree that it would be best if we could run all our experiments on very large datasets (i.e. The Pile, C4). As you mentioned, this would be quite computationally expensive, which was the primary reason we were unable to run these experiments. However, we are able to analyze our main claim (Larger models memorize faster) on the data used to train RoBERTA (see section 4.1), which allows us to consider the under-parameterized regime for all our model scales (because the dataset has close to 39B tokens).
>
> We have added a portion of text *explicitly* addressing the dataset size limitation in section 3 (see L130-132 the updated draft)
>
> W3: Lack of explanation of POS (part of speech) results:
> “The study on parts of speech is especially interesting, however the section lacks a bit on the explanation of the behaviour … It would be interesting/beneficial for the paper to study this angle further even using Wiki Text 103 dataset, to see if there are any correlation with memorization with the frequency of individual POS class.”
>
> A: We agree that the frequency of a particular POS class could aid in memorization of that POS class, especially since previous studies have shown that higher sequence frequency leads to higher sequence memorization [1]. Our intention with this section was just to introduce the idea of analyzing “what is memorized when” through the lens of POS, not necessarily determine why the POS trend occurs, although we hope to analyze this in future work.
>
> ### Questions
> Q1: In L122, the authors seem to refer the BookCorpus as “RoBERTa” dataset. Any specific reason why?
>
> A: If you look at the original RoBERTa paper (See section 3.2 in [2]) there are more datasets than just BookCorpus that are used to train the model, which is why we referred to this dataset as the RoBERTa corpus. This includes BookCorpus, CC-News, OpenWebtext, and Stories.
>
> Q2: Furthermore, BookCorpus which is used to trained RoBERTa model comes in two sizes: 16GB (which is used for RoBERTa-base) and 160GB (which is used for RoBERTa-large). I'm not sure which one is being used here.
>
> A: We used the RoBERTa-large training dataset. The total size is around 160GB.
>
> Q3: What is the model family used in these experiments? Are the results specific to Causal language models or Masked Language Models? Do the authors expect the results to differ based on the class of the model family being used?
>
> A: Our results in section 4 (“Larger models memorize faster”) are for both causal (Figures 1, 3, 4) and masked (Figures 2, 3, 4) language modeling. We also show memorization dynamics for both causal and masked language modeling (on both WikiText103 and the RoBERTa dataset) in the appendix A.1. Apart from those experiments, all other analyses were performed on causal language models. We do not expect the overall trends (with respect to model scale) to differ based on the model family, because the memorization dynamics between model families are similar (see Figure 11/12 in section A.1.).
>
> Q4: In the catastrophic forgetting experiment, the special batch is taken from the validation set of Wiki 103, which is in-distribution with the training set. If I were to choose a special batch being significantly out-of-distribution with respect to the training data, would the same results hold?
>
> A:  We expect that the in-distribution forgetting is an upper bound on the out-of-distribution forgetting, but we expect to see a similar dependence on model size, which is supported by previous work showing that increasing model scale mitigates out-of-distribution forgetting [3]
>
> ### Citations:
> [1] Nicholas Carlini, Daphne Ippolito, Matthew Jagielski, Katherine Lee, Florian Tramer, and Chiyuan Zhang. Quantifying memorization across neural language models. arXiv preprint arXiv:2202.07646, 2022.
>
> [2]: Liu, Yinhan, et al. "Roberta: A robustly optimized bert pretraining approach." arXiv preprint arXiv:1907.11692 (2019).
>
> [3]: Ramasesh, Vinay Venkatesh, Aitor Lewkowycz, and Ethan Dyer. "Effect of scale on catastrophic forgetting in neural networks." International Conference on Learning Representations. 2021.

---

> > ### Comment · Reviewer_YUcS · 2022-08-08
> > **Thank you for your response**
> >
> > Thank you for providing a detailed response to my questions. I'm glad that most of my questions are answered, and I'm happy to improve my original evaluation.

---

### Official Review · Reviewer_3y6g · 2022-07-11

**Rating:** 7
**Confidence:** 4
**Soundness:** 3 good
**Presentation:** 4 excellent
**Contribution:** 3 good

**Summary:**

This paper demonstrates a detailed picture of training data memorization in the process of language modeling. The authors show that larger-sized language models memorize training data faster. Besides, this memorization happens before the overfitting of language modeling. Moreover, specific part-of-speech-tagged tokens are memorized faster like nouns and numbers during training. Lastly, they use the validation set as special insertion batch to test the forgetting mechanism with different model scales.

**Questions:**

The PoS are not of comparable size (e.g. numbers can be considered relative as a closed set compared to nouns, even though they are not actually, and the size of prop. nouns are relatively small compared to the set of nouns). Therefore, the memorization of nouns demonstrated in the experiments may not be as informative as the memorization of more fine-grained categories like numbers.

**Limitations:**

One potential related reference: https://aclanthology.org/2022.tacl-1.1.pdf

**Strengths And Weaknesses:**

Strength
- The experimental results are detailed with the reasonable proposed metrics in data memorization.
- The forgetting identifier experiments are well-designed.
- In general, these fine-grained findings add value to drawing the picture of how and what data are memorized in transformer-based language modeling.

Weakness:
- Some example-based analysis (e.g. the POS-wise breakdown of memorization and the forgetting experiment) would be helpful in further demonstrating the procedure.

---

> ### Author Response · Authors · 2022-08-02
> **RE: Reviewer 3y6g**
>
> Thank you for reviewing our work! We are happy that you found our experiments well-designed and agree with our main points.
>
> ### Weaknesses
>
> W1: Some example-based analysis (e.g. the POS-wise breakdown of memorization and the forgetting experiment) would be helpful in further demonstrating the procedure.
>
> A: We are not entirely sure what you mean by “further demonstrating the procedure” but we are guessing that you mean that you would like to see the example-based analyses rerun with different configurations (i.e. changing hyperparameters)?
>
> For the forgetting experiment, we are able to consider varying model scales up to 13B, but sweeping hyperparameter/model families at this scale is very computationally expensive, so we leave such experiments for future work. Previous work has shown that large scale language models  are relatively robust to variations in learning rate and related parameters [2] (see 3rd paragraph in Section 7 in [2]).
>
> For the POS-wise breakdown of memorization, there is  work [1] that conducts a similar study with smaller model sizes and a different language model family (masked language models). They also find that different parts of speech are learned at different speeds.
>
> For both example-based analyses, our intention was to introduce ways of analyzing how language models memorize training data, rather than to provide a full sweep of configurations (which we leave as future work).
>
> ### Limitations:
> L1:One potential related reference: https://aclanthology.org/2022.tacl-1.1.pdf
>
> A: We agree that this is a relevant citation, and have included in the “Related Works” section of our paper (see L59 in the updated draft).
>
> ### Citations:
> [1]: Cheng-Han Chiang, Sung-Feng Huang, and Hung-yi Lee. 2020. Pretrained Language Model Embryology: The Birth of ALBERT. In Proceedings of the 2020 Conference on Empirical Methods in Natural Language Processing (EMNLP), pages 6813–6828, Online. Association for Computational Linguistics.
>
> [2]: Kaplan, Jared, et al. "Scaling laws for neural language models." arXiv preprint arXiv:2001.08361 (2020).

---

### Official Review · Reviewer_xEvT · 2022-07-11

**Rating:** 6
**Confidence:** 3
**Soundness:** 2 fair
**Presentation:** 3 good
**Contribution:** 3 good

**Summary:**

In this paper, the authors define memorization as any situation in which the model predicts the correct word as maximum probability given its context. They find that larger models memorize faster under this definition. They then study the asymptotic behavior of forgetting the memorized samples, finding that larger models likely never completely forget their memorized samples. They then study memorization paper according to part of speech, suggesting that rare words tend to lead to memorized sequences.

**Questions:**

Why did you pick the particular reference implementations that you picked? Are these considered standard for work like this?

lines 240-242: Does this imply that the special batch is not seen once immediately when introduced, but is seen later on? Or is this implying that the spectral batch is seen repeatedly in each epoch? That was not my understanding initially, en did makes this confusing

line 202: Maybe clarify that this means memorization of which words correspond to a particular part of speech? Or are you claiming that the correct part of speech is consistently selected? Because that should definitely not be referred to as memorization.

187-189, 92-95: Could you rewrite these sentences? I find them very confusing.

**Limitations:**

I didn't feel that the authors were precise enough in their definitions, leading them to use some words, like memorization, in ways that might not reflect what we usually mean when we talk about memorization. I think being more precise in their language would help this paper greatly, as the results that they describe might not influence other notions of memorization.

**Strengths And Weaknesses:**

Strengths:

The question of how memorization occurs over the course of training is an interesting one. I like that they bring in implications for privacy, though I’d like to see more explicit links to the generalization literature.

lines 227-228: As the number of parameters increases, the model forgets less. Seems obvious, but actually really nice to see this demonstrated in such a simple way. In general, some of their results seem intuitive, so when their methods are convincing and well explained, it's satisfying. This sort of work often finds people complaining that these are things that we already know, but if they aren't tested in the current empirical work, this kind of result has a lot of value. The results on forgetting curves and the lower bounds that they exhibit at various scales was convincing and presented a good framing for the phenomenon of forgetting.

Weaknesses:

A broad issue I have is I'm not convinced "memorization" is the appropriate term for the behavior discussed.

It's not clear to me that every time the correct word is predicted as maximum probability, it should be counted as memorization rather than generalised learning. Also, how is this affected by conditions where the same context appears multiple times in the training set with different labels? If every time a certain sequence occurs in training, it always occurs with a particular word, it seems that it's not memorization to indicate that that's the highest probability word.

 What does it mean for a part of speech to be memorized? I don't agree that a part of speech is memorized just because it is predicted correctly in context. There aren't that many parts of speech, so unlike when trying to predict the missing word in a unique sentence, I don't think that it's appropriate to claim that a sequence is memorized just because the predictions made are the correct part of speech. 207 highlights this problem, as it refers to the behavior first as learning and then as memorizing.

209: If the model is just learning to predict that the token is a numeral, and not learning to predict the specific numeral, it’s not clear to me that this is actually going to have privacy implications.

In terms of the uniqueness experiments, I would want to see what the effect of word frequency is in general before considering the role of completely unique words.

174: "classical ML concepts cannot even explain such a memorization trend." Not sure what this means, you should probably talk about the concepts that you feel fail to explain the trend.

I'm not convinced by the experiments that consider memorization relationship with overfitting as though these are two separate phenomena. What if the model is overfitting on a subset of the data, but generalization is improving overall, just not on examples that might be related to that subset? Seems like overfitting by this metric might be an emergent property of having enough memorization occur.

Problems with sparse literature review:

This work is not contextualized in the existing literature on training dynamics in language models, outside of recent literature on general scaling laws. I recommend looking through a variety of work on the area: https://www.semanticscholar.org/search?q=training%20dynamics%20language%20models&sort=relevance

In general, I take issue with the lack of citations to existing work. Is label memorization extending an existing concept? You wouldn't think so, from the paper. There is a mention of spaced repetition from the cognitive science literature, but the authors do not acknowledge that it has also been applied in natural language processing training: https://aclanthology.org/D17-1255/

Many concepts are introduced without any kind of citation to the existing literature: catastrophic forgetting, machine unlearning. Terms like "basin" and "phase transition" are introduced without background, so it's not clear what type of phenomenon each of these phrases is intended to refer to in this context.

Minor:
- 132: "generally monotonically decreasing" just say that it's generally decreasing. It's not monotonically decreasing, and appending "generally" in this context just means "not monotonic".
- 51: \citep should be \citet here

---

> ### Author Response · Authors · 2022-08-02
> **RE: Reviewer xEvT**
>
> Thank you for such an insightful review!  We would like to address your points individually below:
>
> ### Weaknesses:
>
> *W1: A broad issue I have is I'm not convinced "memorization" is the appropriate term for the behavior discussed.*
>
> A: We understand the hesitancy to call what we study in section 4 “memorization” since it can also be seen as accuracy (we highlight this on L95-96). Memorization itself is not well defined in the case of language modeling, with multiple works all defining memorization differently (for example [1] in section 3.1, [2] in section 3.1.1, [3] in section 2.1). The reasons we chose our particular metric:
>
> (1) We are interested in memorization from a privacy/fairness perspective, which means we first need to study the question: if we deploy this model, and someone prompts with a context from training data, will the model output exactly what was in the training data? More importantly, how does the transition from [not outputting training data] => [outputting training data] depend on factors such as scale?
>
> Once we’ve characterized this behavior, we can ask more complicated question such as:
> How robust is memorization of training data: what happens if we change particular words in the sentence, or if we make the input context length longer? Does the model still consistently output the same word?
> What about semantic memorization: what if the model always outputs a label y’ which conveys the same information as y (for example, a synonym)?
>
> While these sorts of experiments more clearly align with the natural concept of memorization, we still need to establish the baseline of how quickly the model exactly outputs training data.
>
> (2) “Memorization” is also sometimes defined as perfectly fitting the training data [7, 8]. However, this definition only works if you consider the final model after training. We are interested in how the model transitions from [not fitting training data at all] => [perfectly fitting training data = “memorization”]. In order to do that, we must use a metric that measures the degree of memorization, and analyze how it evolves over training. Therefore, even if intermediate values of our metric might not be extreme enough to be considered full “memorization,” we still believe it is valid as a measure of the degree of memorization.
>
> *W2: Also, how is this affected by conditions where the same context appears multiple times in the training set with different labels? If every time a certain sequence occurs in training, it always occurs with a particular word, it seems that it's not memorization to indicate that that's the highest probability word.*
>
> A: To address the first question: we believe that if we train the model on examples (c, y) and (c, y’), then the model will become less confident in its prediction when prompted with “c” (although it is unclear which label it will choose).
>
> To address the second sentence: We believe the important factor here is context length.
>
> We agree that there are short contexts with (probably) high sequence frequency, which if consistently outputted, does not necessarily indicate memorization — for example, the sequence “in the” is a very common phrase in natural english that is likely to appear frequently in a dataset. However, there are also short contexts that appear multiple times in WikiText103 such as “The Boat Race” (see [6], section 5.2, in the section title “What causes supercopying”). This is not an extremely common phrase in normal English, so consistently outputting this sequence can be considered more memorization of training data than general learning.
>
> On the other hand, if we have long contexts that the model consistently outputs, even if it is due to high sequence frequency, we believe it is more indicative of memorization. This is because these long sequences are unlikely to be very frequent in general English. As a concrete example, [6] shows that there are high-frequency long contexts (100-grams) in WikiText103 that language models consistently output, which they refer to as “supercopying” (see Section 5.2). They attribute this behavior to high sequence frequency (see Figure 9), and if you inspect Figure 8, you see some examples of these 100-grams, which clearly shows that they are not likely to be frequent in natural English.
>
> Since previous work [1] has shown that short context length sequences are difficult to extract (which implies they are hard to memorize by our metric: our memorization metric implies extractability as defined in [1], so (not extractible) implies (not memorized)), we did not focus extensively on distinguishing different cases within the family of short context-length sequences.

---

> > ### Author Response · Authors · 2022-08-02
> > **RE: Reviewer xEvT (part 2)**
> >
> > *W3: What does it mean for a part of speech to be memorized?... I don't think that it's appropriate to claim that a sequence is memorized just because the predictions made are the correct part of speech.*
> >
> > A: If the model predictions are the correct part of speech, we do not claim that the “sequence is memorized” but that the parts of speech for a sequence are memorized. We agree that predicting the part-of-speech correctly for individual words may not be extreme enough to constitute “memorization.” However, if *all* the parts of speech for a given sequence are consistently predicted correctly, we believe that constitutes “memorizing parts of speech” (especially considering that the average sequence length in WikiText103 is 430.12 tokens). As we mentioned in the answer to W1, we study the transition from [not predicting any parts of speech correctly] => [predicting all parts of speech correctly = “memorizing parts of speech”].
> >
> > *W4: 209: If the model is just learning to predict that the token is a numeral, and not learning to predict the specific numeral, it’s not clear to me that this is actually going to have privacy implications.*
> >
> > A: We see what you are saying and agree that only knowing part-of-speech has limited privacy implications; however, we disagree that it has no privacy implications. If the training set contains a prompt such as:
> >
> > “The password for root is _ _ _ _ _”
> >
> > and the language model has completely memorized parts-of-speech (it can correctly predict part-of-speech), it makes it easier for an attacker to extract the password for `root`  since they know what positions should be numerals.
> >
> > Also, right plot in Figure 7 demonstrates that the model learns to verbatim output nouns/proper nouns/numerals faster than other parts of speech. This has privacy implications, since in this case the training data itself is exposed.
> >
> > In any case, we have tempered the language on that line to reflect that there are “potential privacy implications” and not necessarily direct privacy implications.
> >
> > *W5: In terms of the uniqueness experiments, I would want to see what the effect of word frequency is in general before considering the role of completely unique words.*
> >
> > A: Previous work has extensively shown that memorization increases as word frequency (or sequence frequency) increases [1, 2, 4]. Most related to our experimental setup, [1] systematically increases sequence frequency in the training dataset, and measures the impact on memorization.
> >
> > For masked language models, [1] defines memorization the same way we do (see Section 5.1 in the `Model and dataset` section). Figure 4c in [1] shows that increasing string frequency increases memorization.
> >
> > For causal language models [1] defines a sequence as memorized if (a) it is present in training data and (b) it is extractable from the language model using greedy decoding. This simply extends our definition to consider memorization of entire sequences, rather than individual (context, word) pairs. We see in Figure 1b in [1] that across model sizes, increasing string frequency increases memorization.
> >
> > This motivates the use of completely unique words — if we use non-unique words, then any measured increase in memorization could be attributed to increased frequency of words, as opposed to the model uniquely identifying training examples.
> >
> > *W6: 174: "classical ML concepts cannot even explain such a memorization trend." Not sure what this means, you should probably talk about the concepts that you feel fail to explain the trend.*
> >
> > A: The concepts we had in mind were along the lines of what we tested in the paper (hyperparameters such as learning rate, concepts from learning theory such as overfitting, complexity measures). However, we realized that we are not running experiments ruling out any other concepts, so we decided to remove this sentence to avoid misleading readers. Thank you for pointing out that this sentence was not clear.

---

> > > ### Author Response · Authors · 2022-08-02
> > > **RE: Reviewer xEvT (part 3)**
> > >
> > > *W8: Sparse literature review*
> > >
> > > *(1) This work is not contextualized in the existing literature on training dynamics in language models*
> > >
> > > A: Thank you for pointing this out, we agree we missed a section of relevant work. We have added a paragraph of relevant work on training dynamics in language models in Section 2.
> > >
> > > *(2) Is label memorization extending an existing concept?*
> > >
> > > A: We have added relevant citations when we introduce label memorization in Section 3, as well as a footnote informally describing what “label memorization” is referring to.
> > >
> > > *(3) There is a mention of spaced repetition from the cognitive science literature, but the authors do not acknowledge that it has also been applied in natural language processing training: https://aclanthology.org/D17-1255/*
> > >
> > > A: This is a very relevant citation for our spaced repetition forgetting experiments, thank you for mentioning this! We have added the relevant citation in Section 5 (see L247).
> > >
> > > *(4): Many concepts are introduced without any kind of citation to the existing literature: catastrophic forgetting, machine unlearning*
> > >
> > > A: We define and cite both catastrophic forgetting and machine unlearning in Section 2 (in “Forgetting in Language Models” subsection). See L64-66 for catastrophic forgetting and L72-L73 for machine unlearning.
> > >
> > > *(5): Terms like "basin" and "phase transition" are introduced without background, so it's not clear what type of phenomenon each of these phrases is intended to refer to in this context.*
> > >
> > > A: We agree it would make the paper more clear to precisely define these terms, since they are general terms that many researchers use. We have added clarification for these terms in the updated draft. Note: We decided to remove the mention of “basin” since the described phenomena in Figure 5 does not consistently show long flat regions of T(N, 0.9) as we vary LR; we think “lowest point on the curve” better describes what we mean and is more self-explanatory than “basin.”
> > >
> > > ### Minor Comments:
> > > Thank you for pointing out these details, we have fixed them in the updated draft.
> > >
> > > ### Questions:
> > >
> > > *Q1: Why did you pick the particular reference implementations that you picked? Are these considered standard for work like this?*
> > >
> > > A: For training models at the 1B+ scale, there are not a lot of open-source reference implementations available. We chose our particular implementations/frameworks because they have open sourced models [9] and are consistent — both in configuration (see Table 1 and Section 2.1 in [9]) and performance (see Section 3 and Figure 4 in [9]) — with standard large-scale language model implementations such as GPT-3.
> > >
> > > *Q2: lines 240-242: Does this imply that the special batch is not seen once immediately when introduced, but is seen later on? Or is this implying that the spectral batch is seen repeatedly in each epoch? That was not my understanding initially, en did makes this confusing*
> > >
> > > A: In L240-242, the special batch is only seen once immediately when introduced. We have updated the draft on L228 to state this more explicitly, thank you for pointing out that this is confusing.
> > >
> > > *Q3: line 202: Maybe clarify that this means memorization of which words correspond to a particular part of speech? Or are you claiming that the correct part of speech is consistently selected? Because that should definitely not be referred to as memorization.*
> > >
> > > A: With the metric R(p) on L202 (or L207 in the updated draft), we are claiming to track memorization of which words correspond to a particular part of speech. We have clarified it in the updated draft on L207, thanks for noting that this phrasing is confusing.
> > >
> > > *Q4: 187-189, 92-95: Could you rewrite these sentences? I find them very confusing.*
> > >
> > > A: We have updated these portions in the draft (hopefully they are more clear now, but please let us know if that is not the case).

---

> > > > ### Author Response · Authors · 2022-08-02
> > > > **RE: Reviewer xEvT (part 4)**
> > > >
> > > > ### Citations:
> > > > [1] Nicholas Carlini, Daphne Ippolito, Matthew Jagielski, Katherine Lee, Florian Tramer, and Chiyuan Zhang. Quantifying memorization across neural language models. arXiv preprint arXiv:2202.07646, 2022.
> > > >
> > > > [2] Nicholas Carlini, Florian Tramer, Eric Wallace, Matthew Jagielski, Ariel Herbert-Voss, Katherine Lee, Adam Roberts, Tom Brown, Dawn Song, Ulfar Erlingsson, et al. Extracting training data from large language models. In 30th USENIX Security Symposium (USENIX Security 21), pages 2633–2650, 2021.
> > > >
> > > > [3] Chiyuan Zhang, Daphne Ippolito, Katherine Lee, Matthew Jagielski, Florian Tramèr, and Nicholas Carlini. Counterfactual Memorization in Neural Language Models. arXiv:2112.12938 [cs], December 2021. arXiv: 2112.12938 version: 1.
> > > >
> > > > [4]: Lee, K., Ippolito, D., Nystrom, A., Zhang, C., Eck, D., Callison-Burch, C., and Carlini, N. (2021). Deduplicating training data makes language models better. CoRR, abs/2107.06499
> > > >
> > > > [5]: Merity, Stephen, et al. "Pointer sentinel mixture models." arXiv preprint arXiv:1609.07843 (2016).
> > > >
> > > > [6]: McCoy, R. Thomas, et al. "How much do language models copy from their training data? evaluating linguistic novelty in text generation using raven." arXiv preprint arXiv:2111.09509 (2021).
> > > >
> > > > [7]: Chiyuan Zhang, Samy Bengio, Moritz Hardt, Benjamin Recht, and Oriol Vinyals. Understanding deep learning requires rethinking generalization. arXiv:1611.03530 [cs], February 2017. URL http://arxiv.org/abs/1611.03530. arXiv: 1611.03530
> > > >
> > > > [8]: Pondenkandath, Vinaychandran, et al. "Leveraging random label memorization for unsupervised pre-training." arXiv preprint arXiv:1811.01640 (2018).
> > > >
> > > > [9]: Zhang, Susan, et al. "Opt: Open pre-trained transformer language models." arXiv preprint arXiv:2205.01068 (2022).

---

> > > > > ### Author Response · Authors · 2022-08-02
> > > > > **RE: Reviewer xEvT (part 5)**
> > > > >
> > > > >
> > > > > *W7: I'm not convinced by the experiments that consider memorization relationship with overfitting as though these are two separate phenomena. What if the model is overfitting on a subset of the data, but generalization is improving overall, just not on examples that might be related to that subset? Seems like overfitting by this metric might be an emergent property of having enough memorization occur.*
> > > > >
> > > > > A: In practice, overfitting is defined with respect to a fixed train/test split (if we do not have out-of-sample data to evaluate model performance on, we have no way of knowing that the model is fitting noise in the train set). Our goal in this section was to see if larger models memorize faster, because they overfit faster with respect to the given train/test split. In other words, we are considering “memorization on the train/test split” and “overfitting on the train/test split” as separate phenomena, not necessarily “memorization” and “overfitting.”
> > > > >
> > > > > We agree that in non i.i.d. settings, the model could be overfitting on subsets of data while generalization improves. However, Wikitext103 is relatively i.i.d. because all data is collected from high quality, factual, and neutral Wikipedia articles (see Section 4.3 in [5]). Therefore, we do not believe the situation you describe occurs very frequently, although confirming this experimentally is intractable (would require evaluating over all possible subsets of training data).
> > > > >
> > > > > Nevertheless, we see your point and are hoping to re-run this experiment over different train/test splits and include results in the appendix in the final version (we are unable to do so in the short timeline of a week, due to the scale of models we consider).

---

### Meta-Review · Area_Chair_nahA · 2022-08-25

**Recommendation:** Accept
**Confidence:** Certain

**Metareview:**

This paper studies the underlying training and memorization dynamics of very large language models. The main take aways are that larger-sized language models memorize training data faster, and that this memorization happens before the overfitting of language modeling. Tokens with certain part-of-speech tags (nouns, numerals) seem to be memorized faster during training.

Overall, most reviewers feel positively about this paper, agreeing that it tackles an important problem and that it provides a solid contribution. The experimental results are detailed and use reasonable metrics for data memorization, including the forgetting identifier experiments. Some of the weaknesses that have been pointed out (e.g. regarding the significance of the part-of-speech tags experiment, clarifying the criteria for memorization, etc.) seem to have been well addressed during the author response. Therefore, I recommend acceptance.



**Award:**

No

---

### Decision · Program_Chairs · 2022-09-14

Accept